# Sub-Antarctic Freshwater Invertebrate Thermal Tolerances: An Assessment of Critical Thermal Limits and Behavioral Responses

**DOI:** 10.3390/insects11020102

**Published:** 2020-02-04

**Authors:** Javier Rendoll-Cárcamo, Tamara Contador, Peter Convey, James Kennedy

**Affiliations:** 1Wankara Sub-Antarctic and Antarctic Freshwater Ecosystems Laboratory, Sub-Antarctic Biocultural Conservation Program, Universidad de Magallanes, Puerto Williams, Teniente Muñoz 166, Chile; tamara.contador@umag.cl (T.C.); James.Kennedy@unt.edu (J.K.); 2Institute of Ecology and Biodiversity, Universidad de Chile, Santiago, Las Palmeras 3425, Chile; 3Antarctic and Sub-Antarctic Sciences Ph.D. Program, Universidad de Magallanes, Punta Arenas, Avenida Bulnes 01855, Chile; 4Núcleo Milenio de Salmónidos Invasores, INVASAL, Iniciativa Científica Milenio, ICM, Núcleo Científico Milenio, Concepción, Casilla 160-C, Chile; 5British Antarctic Survey, NERC, Cambridge CB3 0ET, UK; pcon@bas.ac.uk; 6Department of Biological Sciences, University of North Texas, 1511W Sycamore, Denton, TX 76201, USA

**Keywords:** critical thermal limits, ecophysiology, elevation, freshwater macroinvertebrates, restricted distribution

## Abstract

Physiological thermal limits of organisms are linked to their geographic distribution. The assessment of such limits can provide valuable insights when monitoring for environmental thermal alterations. Using the dynamic critical thermal method (CTM), we assessed the upper (CT_max_) and lower (CT_min_) thermal limits of three freshwater macroinvertebrate taxa with restricted low elevation distribution (20 m a.s.l.) and three taxa restricted to upper elevations (480 and 700 m a.s.l.) in the Magellanic sub-Antarctic ecoregion of southern Chile. In general terms, macroinvertebrates restricted to lower altitudinal ranges possess a broader thermal tolerance than those restricted to higher elevations. Upper and lower thermal limits are significantly different between taxa throughout the altitudinal gradient. Data presented here suggest that freshwater macroinvertebrates restricted to upper altitudinal ranges may be useful indicators of thermal alteration in their habitats, due to their relatively low tolerance to increasing temperatures and the ease with which behavioral responses can be detected.

## 1. Introduction

Understanding the mechanisms by which species distributions change along geographical gradients has been a key tool in biogeography, ecology, and evolution [1], and in the development of macroecological theories. Latitudinal and altitudinal gradients generally encompass several environmentally associated variables and can be considered ecological and evolutionary equivalents in terms of their influences on species adaptations [2]. The variations in and expression of species physiological traits are a direct response to the environment [2,3]. The “climate variability hypothesis” (CVH) predicts that more variable climates may select for organisms with broader thermal tolerances, whereas less variable (stable) climates select for narrower thermal tolerances [4,5,6,7,8]. High latitude environments are often characterized by broad thermal variation while low latitude or tropical environments have a narrow thermal regime. Organisms inhabiting high latitudes are therefore expected to express broader physiological thermal tolerances, especially to withstand cold temperatures [9,10]. On the other hand, tropical species will show relatively narrow thermal tolerance ranges that are appropriate to the local environmental thermal variation [8,10].

Janzen [6] considered the influence of elevation on physiological thermal ranges in an analogous fashion. In tropical mountains, freshwater ecosystems have a narrow annual temperature range, leading to species with narrow thermal tolerances. In contrast, in temperate mountains, seasonal changes in temperature and variation across the elevation gradient should select for species that can tolerate a wide range of temperatures [8]. This relationship has received attention lately [2,8,11,12,13,14]. Elevation gradients are also relevant in terms of changes in environmental conditions, such as air and water temperature, oxygen availability, snow or ice cover, and the availability of different habitat types, among other factors [15]. Water temperature has been one of the most studied abiotic factors, being a relevant ecological driver in aquatic ecosystems [16], and influencing various aspects of the biology of organisms [17]. Hence, an understanding of the thermal sensitivity of organisms provides a key indicator to determine early changes in water temperature. Sensitive organisms can provide insight as “sentinels” of environmental alterations [17].

Critical thermal limits can provide useful information on temperature influences on physiological and ecological responses that impact organism survival and species distributions [18,19,20]. Dynamic non-lethal experimental approaches involve increasing or decreasing the temperature at a constant rate to a predefined non-lethal point. One of the most commonly used methods is the critical thermal method (CTM), used to determine the critical thermal endpoint (CTE), expressed as the critical thermal maximum (CTmax) or minimum (CTmin) [17]. These limits represent the endpoints of a performance curve, expressed as the physiological response to a change in an environmental variable, in this case, the temperature [20]. However, there are relevant studies of temperate and tropical freshwater ectotherms [8,21,22,23,24] while sub-Antarctic regions have received very little attention [25].

The Magellanic sub-Antarctic ecoregion is located in southern South America between latitudes 40 and 60° S. It is considered an ideal natural laboratory to assess biotic responses to environmental changes because its geographical setting allows for unpolluted water courses and bodies with minimum confounding influences [26,27]. Additionally, the global land:ocean ratio at these latitudes (2:98) generates a strongly oceanic-buffered climate at lower altitudes, but this effect rapidly decreases with elevation, resulting in a steep thermal gradient with associated changes in landscape structure and biotic turnover [27,28]. In the south of this ecoregion lies Navarino Island (55° S), whose coastal mountain ranges demonstrate these steep gradients over short geographic ranges (~1000 m in altitude), with clear changes in air and water temperatures and vegetation [27].

With this background, the main objectives of the current study were to (i) assess the CTmax and CTmin of selected representative Magellanic sub-Antarctic freshwater macroinvertebrates whose altitudinal distributions are restricted to specific elevation ranges, (ii) compare their thermal tolerance ranges and habitat thermal breadths, and (iii) assess their behavioral responses to warming in order to identify their sensitivity and suitability as sentinel organisms for monitoring environmental thermal changes.

## 2. Materials and Methods

### 2.1. Area of Study

The Magellanic sub-Antarctic ecoregion, located in southern Chile, is characterized by watersheds with acute environmental clines, diverse vegetation profiles, and a wide variety of habitats and microhabitats over a short elevation [27,28]. This ecoregion is part of the South American forest biome, harboring the largest forest and wetland areas of the Southern Hemisphere [26]. Embedded within the ecoregion is the Cape Horn Biosphere Reserve (CHBR; Figure 1), an area designated to protect the ecoregion from the pressures of global change [29]. Furthermore, this region contains some of the world’s cleanest rainwater, as it is located to the south of the typical tracks of industrial-polluted wind currents [30,31,32]. Navarino Island lies in the southern part of this ecoregion. Freshwater invertebrates (amphipods, diving beetles, midges, stoneflies, water boatmen, planarians) were collected from lagoons located within the Róbalo River watershed, which provides domestic water supply to Puerto Williams, the southernmost town in the world (Figure 1).

### 2.2. Site Description

At the coastline, the average air annual temperature is 5.7 °C, while above the tree line (~500–600 m a.s.l.) it is 0 °C [33]. The average annual temperature of one of the rivers within Navarino Island is 5.7 °C at 120 m a.s.l. and only 1 °C at 586 m a.s.l. [34]. The temperature decreases by approximately 1 °C per 100 m of elevation increase, compared to the global average decrease of 0.6 °C per 100 m of elevation [35].

### 2.3. Water Temperature

Temperature loggers (HOBO^®^ model U22 Water Temp Pro Version 2) were installed at a 20 to 30 cm depth in the littoral areas of each lagoon (Castor, el Salto and Bandera, 20, 480 and 700 m. a.s.l., respectively, Figure 1) from which macroinvertebrates were collected over the entire study period (from February 2015 to March 2016). These recorded the water temperature every 4 h. The monthly and annual maximum, minimum, and mean temperatures were calculated from these data.

### 2.4. Collection of Study Organisms

Sampling was carried out in autumn 2015 and 2016 (March and April) to avoid immature stages and allow the collection of identifiable adults, advanced larval stages, or both. Organisms inhabiting medium and high elevation were collected in March because of accessibility to sampling sites while a lower elevation site was surveyed in April, this difference being driven by. The main reason of this sampling lapse is the accessibility to sites and prevalent harsh weather conditions. We followed key considerations for the selection of taxa: organisms collected from the same location (to avoid variability due to the collection site), available in sufficient numbers (relatively abundant and ease of collection), practicability of identification, and suitability for accurately measuring the point of thermal reactivity (PTR) and thermal critical end point (CTE). The PTR is defined as the point at which an organism exhibits signs of thermal stress (i.e., obvious change in body movement, swimming, or crawling capability) while, at the CTE, locomotor functions and activity becomes disorganized to the extent that the organism loses the ability to escape from the conditions, leading to death [17,36]. This last step was achieved through an initial test to observe and identify the PTR and the CTE. We selected six macroinvertebrate taxa inhabiting lagoons from the watershed, three restricted to low elevation (20 m a.s.l.), one to a medium elevation lagoon (480 m a.s.l.), and two to high elevation (700 m a.s.l.) (Table 1). The invertebrates were collected with a D-frame net (150 µm mesh), in addition to hand collection with soft forceps to avoid injury to the organisms. Collected specimens were transferred to a cool box containing site water and transported to the laboratory where they were held in aquaria with an air pump. Before experiments, invertebrates were kept at 5 ± 1 °C (approximate water temperature at collection site) for 24 h. No mortality was recorded during this 24 h period undercontrol conditions, suggesting all collected invertebrates were ‘healthy’ and would have continued to survive this control phase for the duration of the CT_min_ and CT_max_ experiments, which lasted a maximum of 6 h. Temperature was also recorded *in situ* with a multimeter sensor (Conductivity pH TDS Hanna Tester HI98130). After completion of the experiments, the invertebrates were returned to their collection locations.

### 2.5. Critical Thermal Limits

For CTmax assessment, invertebrates were placed in an experimental chamber (15 × 10 × 7 cm) and submerged in a thermoregulated digital water bath (Lab. Companion RW-0525G, Figure 2). Two to six individuals of each taxon were assessed per trial (*n* = 3–4 per taxa), all in the same chamber. An air pump was placed in each experimental chamber to maintain the oxygen saturation above 65% during trials, due to its relevance for thermal tolerance [37]. The first 60 min in the chamber was a control phase, with water at the temperature of the place of collection (5 °C). Subsequently, the temperature was increased at a constant rate (0.14 °C min^−1^) through the experimental phase. The rate of increase aimed to be rapid enough to avoid acclimatization but slow enough to ensure that any reaction observed was a response to the rise in temperature. The reaction of the organisms was evaluated by observing their behavioral responses [36]. Observation of each individual’s reaction took between 30 s to 1 min (i.e., detection of antennal movement, gill, leg, or body movement). When an individual exhibited signs of reaching its CTE (e.g., loss of swimming capacity), the temperature was recorded and it was removed from the experimental chamber and returned to the aquarium at the starting temperature. Only individuals that recovered from the experiment were included in the results (recovery time for CT_max_ trials was 1 h). When 50% (CT_50_) of the individuals showed a response, the trial was terminated [37]. Recovery was assessed by placing each individual in an oxygenated aquarium.

In a similar procedure, the CT_min_ assessment was carried out in experimental chambers submerged in the thermoregulated digital water bath, with an air pump (Figure 2). Each trial had a control phase of 60 min, then a lapse of 15 to 20 min of a temperature decrease (1 °C per trial), and 60 min of exposure to the experimental temperature. Organisms exposed to cooling had a 24 h time frame for recovery. The invertebrates were initially exposed to the collection site temperature, which was subsequently lowered to 0 °C at a constant rate (0.03 °C min^−1^). When 0 °C was reached, the temperature was held for 1 h, and then lowered to −1 °C (0.03 °C min^−1^). Once this target temperature was reached, it was held for 1 h. Ice formation in the chamber was recorded at −1 °C, but the chamber was never frozen completely. After the experiment, individuals were placed in Petri dishes with water at 5 °C (collection site temperature), and allowed to recover for 24 h. The recovered organisms were then placed in the experimental chamber, where the temperature was reduced by 1 °C from the previous test temperature. Only individuals that recovered and exhibited normal locomotive functions (antennal, gill, leg, or body movement) were included in the analyses. When 50% (CT_50_) of the individuals failed to recover, the experiment was terminated. Both experiments were monitored using a 4-channel HOBO^®^ data logging thermocouple (UX120-014M) attached to the thermoregulated bath. Experimental procedures were performed under Bioethics resolution n° 80/CEC/2018 from the University of Magallanes Bioethics committee.

### 2.6. Assessment of Suitability and Thermal Sensitivity Ranks

Suitability ranks (SRs) and the thermal sensitivity ranks (TSRs) were assigned to each of the taxa to compare their behavioral responses to increasing temperature. Both ranks are an adaptation of the method proposed by Dallas and Rivers-Moore (2012). The suitability rank for each taxon was evaluated using the following scale: 1 = very adequate, 2 = adequate with limitations, and 3 = inadequate. Suitability is based on the ease with which the behavioral responses of PTR and CTE are identified, size and age of organisms, and availability in nature. The rank of thermal sensitivity was based on the average of the maximum critical thermal maximum of each taxon, where 1 = very sensitive (≤25 °C), 2 = moderately sensitive (≤30 °C), and 3 = not very sensitive (≥30 °C).

### 2.7. Statistical Analyses

The data were tested for normality (Shapiro–Wilk’s test), and homogeneity of variances (Hartley´s F_max_ ratio). As the data were not normally distributed, a Kruskal–Wallis rank analysis for unbalanced designs was performed. When statistical significance was obtained, further non-parametric comparisons were made using the Wilcoxon method adjusted for multiple comparisons with Holm correction. Regression analyses with linear fit were used to examine the relationship between CT_max_, CT_min_, and thermal breath with elevation. The significance level was set at α = 0.05. Analyses were performed using R version 3.6.1. (R Development Core Team 2019).

## 3. Results

### 3.1. Water Temperature Variability

Using the data retrieved from the loggers, monthly temperature profiles were created for each collection site. At CL (20 m. a.s.l.), the average maximum temperature was recorded in December 2015 (20.6 °C) and the minimum in July 2015 (0.1 °C) while the lagoon surface remained frozen from mid-May to mid-August (Figure 3). At ESL (480 m. a.s.l.), the average maximum temperature was registered in February 2015 (14.2 °C) while the minimum temperature was recorded (0.4 °C) in July of the same year (Figure 3). The high Andean lagoon, BL (700 m. a.s.l.), remained frozen for seven months, and the minimum and maximum temperatures were both recorded in February of 2016 (−1.6 and 14.7 °C, respectively).

### 3.2. Critical Thermal Maxima and Minima of Macroinvertebrates Restricted to Low Elevation

All macroinvertebrate taxa generally showed a relatively wide thermal tolerance range (Figure 4), with CT_max_ being broader than the lagoon temperature range. All species had some tolerance of sub-zero temperatures. The resulting thermal range for the dytiscid *Lancetes angusticollis* was 41.7 °C, between 37.6 (CT_max_) and −4.1 °C (CT_min_), the widest tolerance range of amongst the studied taxa. The water boatman *Sigara* thermal range was 38.6 °C (34.7 to −3.9 °C). Hyalellid amphipods (*Hyalella* sp.) had a range of 35.8 °C (32.4 to −3.4 °C) (Table 2). All organisms (including those that did not recover from experimental exposure) survived sub-zero temperatures and freezing conditions during the experiment.

### 3.3. Critical Thermal Maxima and Minima of Macroinvertebrates Restricted to High Elevation

High-elevation-restricted macroinvertebrate taxa showed narrower thermal ranges compared to those from low elevation. The thermal tolerance range was 28.9 °C for dugesiid planarians, 32.0 °C for the stonefly *Aubertoperla kuscheli*, and 25.2 °C for tanypodinae midge larvae. The latter had the narrowest thermal range of the studied taxa (Figure 4, Table 2).

The results of the Kruskal–Wallis analyses and linear regression fit showed significant differences in the CT_max_ of the studied taxa (H = 114.09, *p* < 0.00001, R^2^ = 0.6917), indicating a marked decrease associated with an elevation increase. Differences were also found in taxa CT_min_ (H = 56.272, *p* < 0.0043, R^2^ = 0.3257, Table 3), particularly between higher elevation taxa (at 700 m. a.s.l. the stonefly *A. kuscheli* and the tanypodine midge) with the other four taxa (at 20 and 480 m. a.s.l.) (Figure 5a), yet not detected by the Wilcoxon test adjusted with Holm correction (Appendix A). Thermal range differences were also detected between taxa restricted to low and high elevation (H = 92.736, *p* < 0.0001, R^2^ = 0.7337, Figure 5b, Table 3). In some cases, the thermal range was twice the temperature range experienced in the environment (Figure 6).

### 3.4. Behavioral Responses, Suitability, and Thermal Sensitivity Ranks

Behavioral responses to the increased temperature varied slightly among the studied taxa. In general, responses included “normal” mobility in the early stages (i.e., swimming, crawling), followed by an abrupt increase in mobility (PTR) and then increasing immobility influencing swimming ability when approaching the final stages of the experiment (CTE). Detailed information on behavioral responses (PTR, CTE, SR, and TSR) is given in Table 4. Macroinvertebrates restricted to high elevation had a narrower thermal tolerance range, so they were assigned to the suitable and thermally sensitive category (Table 4).

## 4. Discussion

The critical thermal method is an effective tool to assess the relative thermal tolerance of freshwater macroinvertebrates. This method also offers the possibility of monitoring behavioral responses, which, in turn, are useful to identify potential bioindicators of thermal alterations [17]. The results obtained here showed significant differences between the critical thermal limits (CT_max_ and CT_min_) of different macroinvertebrate taxa inhabiting the same watershed at different elevations. In particular, the CT_max_ decreased for upper elevation macroinvertebrate taxa when compared to those from lower elevations. While CT_min_ did not show large variation, it was significantly lower for higher elevation macroinvertebrates (Figure 4 and Figure 5, Table 2 and Table 3). Despite the current study lacking closely related invertebrate taxa to compare across the gradient, we identified a consistent underlying pattern of broader thermal tolerance limits in taxa from more variable environments. These results are consistent with other studies [8], and also with Janzen´s climate variability hypotheses predictions [6], more variable climates may select for organisms with broader thermal tolerances, whereas less variable climates select for narrower thermal tolerances [4]. Nonetheless, comparison among families or genera with phylogenetic or physiological affinities, acclimation effects, intensity, and duration of thermal stress, and robust analyses are necessary to improve our understanding on thermal ecology [21].

Midges (Chironomidae: Tanypodinae) had the lowest thermal tolerance range and also the lowest thermal limits while the diving beetle *Lancetes angusticollis* had the broadest thermal tolerance (Figure 4 and Figure 5, Table 2 and Table 3). This species, along with the other two taxa inhabiting low elevation lagoons, can be observed throughout the year, even when ice is present. Additionally, the broader thermal breadth of the studied taxa when compared to the environmental temperature range suggests that, in some cases, these organisms are able to tolerate temperatures well above the maxima recorded for their habitats. These relatively high tolerances might be an adaptation to the annual environmental conditions (long periods of ice cover and sunlight in winter, and warm exposure with an increased photoperiod in summer). In particular, we found that the stonefly *Aubertoperla kuscheli* had a thermal range of 32.1 °C, which is double the thermal range of Bandera Lagoon (700 m a.s.l.). It is also notable that this is the only stonefly present in lagoons in the Róbalo watershed [28].

During the experimental observations, we recorded that stoneflies and planarians respond with stirred movements to ice formation into the chamber, probably because of ice-relationinjuries. These behavioral responses to cold in organisms restricted to high elevations, in addition to their common habitat structure (mosses and submerged vegetation), suggests that they attempt to avoid contact with ice rather than tolerating it. Suren [38] and Suren and Winterbourn [39] noted the role of bryophytes as a refuge and nursery for freshwater macroinvertebrates in habitats subject to harsh environmental conditions. Aquatic and/or semi-submerged bryophytes can provide shelter for the development of immature stages, protection against predators, and may themselves be a food source [39].

The critical limits identified, in addition to the behavioral responses, suggest that temperature may act as a distribution barrier. However, other factors that can be of biotic (e.g., predation, exclusion by competition, etc.) or abiotic origin (e.g., dissolved oxygen, resource availability, etc.), might confound these observations. Studies focused on assessing the long-term thermal regimes in addition to other abiotic variables are necessary to understand how variation in these factors, and their combined effects, influences the biology of freshwater fauna [19]. Shah et al. [8] compared the thermal ranges and tolerances of tropical and temperate aquatic insects, both at latitudinal and altitudinal scales. The CTmax of aquatic insects from temperate regions (Rocky Mountains) were similar to those measured in the sub-Antarctic invertebrates assessed in the present work.

There is a considerable literature on methodological protocols and sources of variation in the determination of critical thermal limits [19,40]. The main sources of variation are acclimation and the rate of temperature change. Recording collection site water temperature is helpful to avoid distortion in thermal limit assessments [17]. The effect of different acclimation temperatures on thermal limits has been addressed in South African amphipods [17,41], different families of Coleoptera [19], and also in sub-Antarctic spiders [25]. Different rates of temperature change can increase or decrease the measured upper and lower thermal limits during experiments, as shown for *Tenebrio molitor* (Tenebrionidae) and *Cyrtobagous salviniae* (Curculionidae) [19]. A further source of variation is the size/age of the organisms examined, but with more complex outcomes, with positive [42], negative [43], or no effects [36] being reported. In the current study, the potential for this source of error was minimized by collecting only adult or late juvenile stage individuals for each test.

## 5. Conclusions

Several considerations should be taken into account when addressing physiological tolerances, including the use of control and acclimation trials, exposure intensity and duration, etc. in an attempt to provide stronger evidence that the observed effects are due to treatments. The predictions of the CVH [6] are consistent with our data, but a robust test of this hypothesis will require further study and analyses. Comparisons between families or genera, further trials, and different treatments are needed in order to improve support for the CVH predictions. Additionally, the potential sources of variation in the thermal ecology studies must be considered, including a temporal dimension and the methodological context on which limits are estimated and then extrapolated into natural settings [21]. New questions arise regarding the determination of thermal limits: Do tolerance ranges of a species inhabiting different watersheds differ? How influential are the thermal regimes of other aquatic systems (including streams)? How sensitive are high elevation organisms to longer periods of exposure to thermal alteration? Addressing such questions will be required to give insight into how this remote southern biota will respond under future global environmental change scenarios.

## Figures and Tables

**Figure 1 insects-11-00102-f001:**
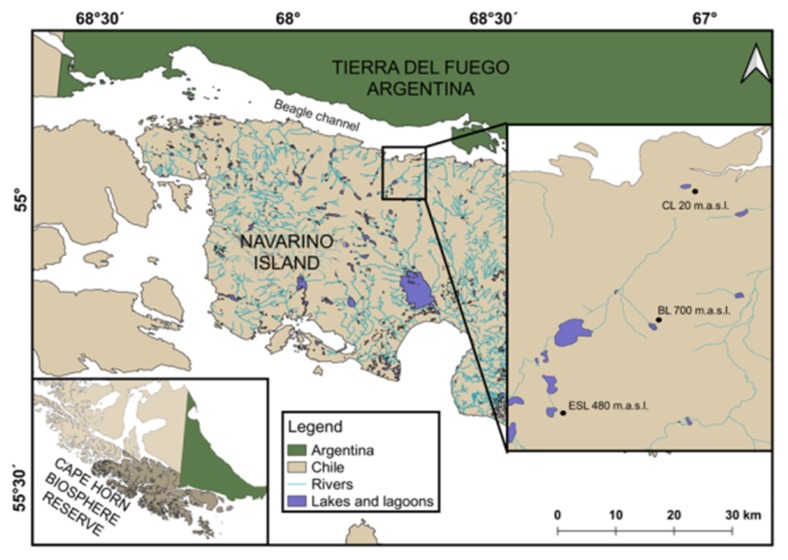
Cape Horn Biosphere Reserve and collection sites on Navarino Island. Collection sites within the Róbalo river watershed are expanded in the inset. CL, Castor Lagoon; ESL, el Salto Lagoon; BL, Bandera Lagoon.

**Figure 2 insects-11-00102-f002:**
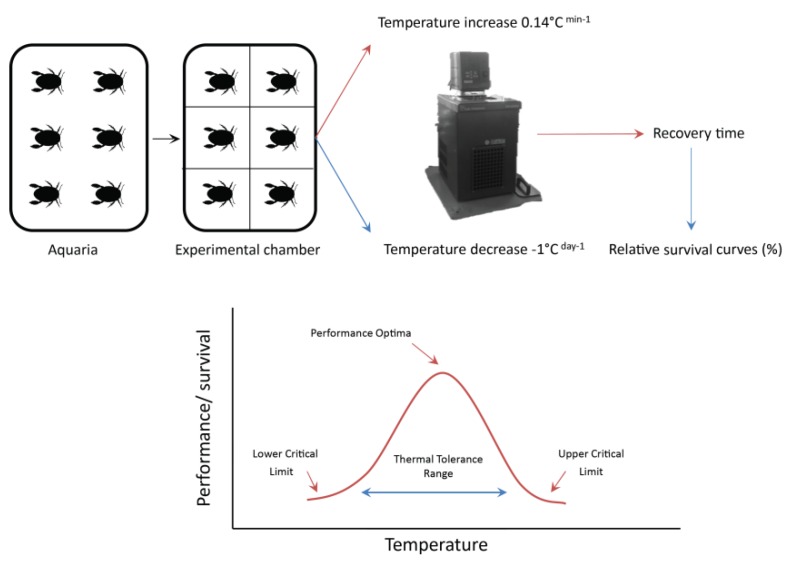
Conceptual scheme of the critical thermal limits determination in the present study.

**Figure 3 insects-11-00102-f003:**
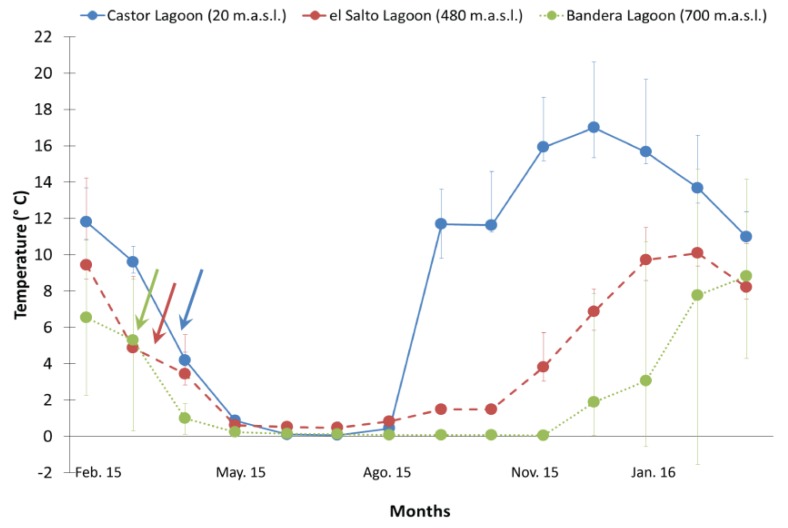
Monthly mean temperatures (±SD) retrieved from loggers installed in lagoons at different elevations in the Róbalo River watershed. Color lines indicate different lagoons, and colored arrows indicate the time of collection at each elevation.

**Figure 4 insects-11-00102-f004:**
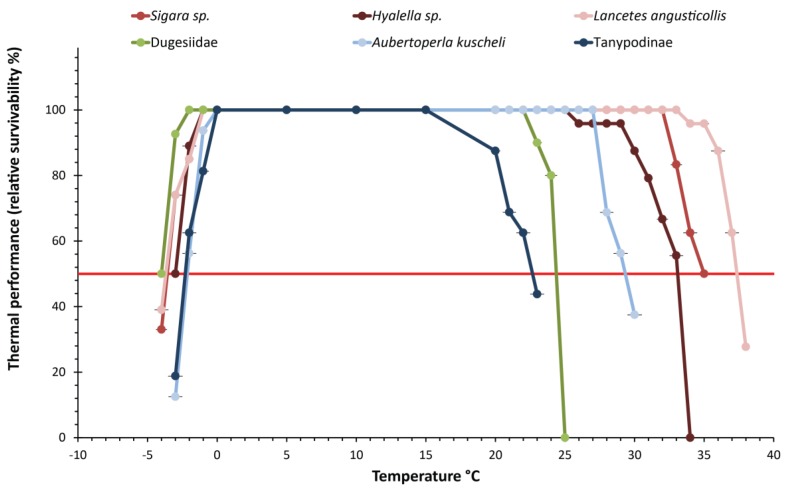
Thermal performance curves (±SD) of studied macroinvertebrate taxa. Red-colored lines indicate taxa restricted to low elevation lagoons, the green-colored line indicate taxa at a medium elevation, and blue-colored lines represent taxa restricted to high elevation lagoons from the Róbalo watershed, Navarino Island, southern Chile. The red line at 50% relative survival denotes CT_50_.

**Figure 5 insects-11-00102-f005:**
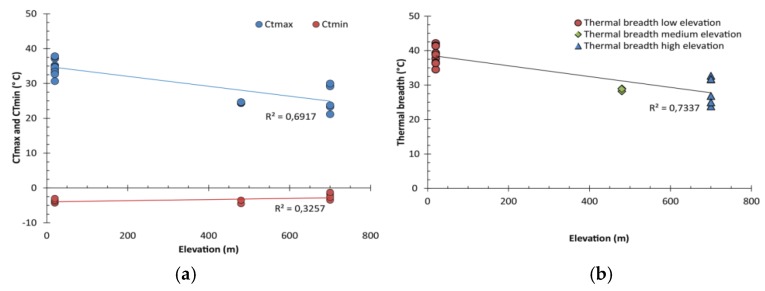
Critical thermal limits and thermal ranges of the studied taxa across the altitudinal gradient of the Róbalo watershed; (**a**) CTmax declined with increasing elevation while CTmin varied little; (**b**) the thermal range also declined significantly with increasing elevation. Linear regression values (R^2^) are shown.

**Figure 6 insects-11-00102-f006:**
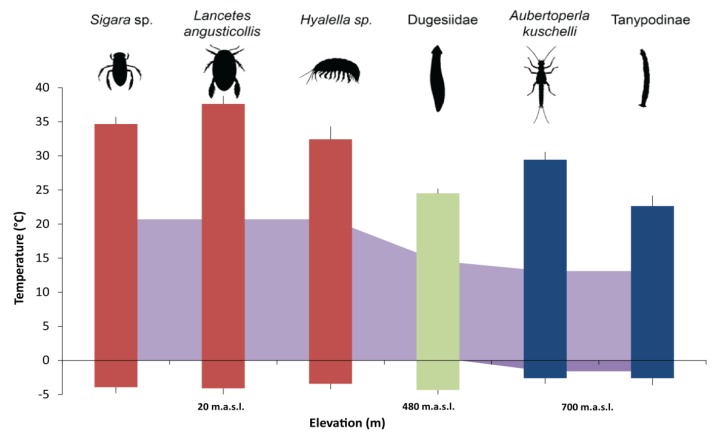
Thermal ranges of the studied taxa (±SD) compared with those of the different Róbalo watershed lagoons temperature range (purple shading in the background). Taxa restricted to a particular elevation are represented by colored bars (red = 20 m a.s.l., green = 480 m a.s.l., blue = 700 m a.s.l.).

**Table 1 insects-11-00102-t001:** Number of individuals per tested taxon and their collection sites within the Róbalo river watershed on Navarino Island, southern Chile. CL = Castor Lagoon, ESL = el Salto Lagoon, BL = Bandera Lagoon.

Macroinvertebrate Taxa	Taxon Elevation Range	*n* CTmax	*n* CTmin	Site	Lat/Long	Elevation (m a.s.l.)
*Lancetes angusticollis* (Dytiscidae)	Broad (from 20 m to 380 m a.s.l.)	24	18	CL	54°56′19′′ S/67°38′15′′ W	20
*Sigara* sp. (Corixidae)	Narrow (only near sea level)	24	18	CL	54°56′19′′ S/67°38′15′′ W	20
*Hyalella* sp. (Hyalellidae)	Family with broad range, morpho-species narrow range.	24	18	CL	54°56′19′′ S/67°38′15′′ W	20
Dugesiidae (Tricladida)	Narrow (only found at medium elevation)	18	14	ESL	54°59′26′′ S/67°40′56′′ W	480
*Aubertoperla kuscheli* (Gripopterygidae)	Narrow (only found at high elevation)	16	16	BL	54°58′26′′ S/67°38′41′′ W	700
Tanypodinae (Chironomidae)	Family with broad range, morpho-species narrow range.	16	16	BL	54°58′26′′ S/67°38′41′′ W	700

**Table 2 insects-11-00102-t002:** CT_max_, CT_min_, and thermal breadth values of the studied macroinvertebrate taxa, (±SD). Values are given in °C.

Macroinvertebrate Taxa	Mean CT_max_	Mean CT_min_	Mean Thermal Breadth
*Sigara* sp. (Corixidae)	34.7 (±1.05)	−3.9 (±0.87)	38.6 (±1.24)
*Lancetes angusticollis* (Dytiscidae)	37.6 (±1.17)	−4.1 (±0.90)	41.7 (±1.52)
*Hyalella* sp. (Hyalellidae)	32.4 (±1.91)	−3.4 (±0.78)	35.9 (±2.59)
Dugesiidae (Tricladida)	24.5 (±0.71)	−4.4 (±0.63)	28.9 (±1.07)
*Aubertoperla kuscheli* (Gripopterygidae)	29.4 (±1.15)	−2.6 (±0.81)	32.1 (±1.29)
Tanypodinae (Chironomidae)	22.6 (±1.54)	−2.6 (±1.02)	25.3 (±1.69)

**Table 3 insects-11-00102-t003:** Details of Kruskal–Wallis analyses (α = 0.05) for CT_max_, CT_min_, and thermal ranges of the six studied macroinvertebrate taxa from the Róbalo watershed lagoons, Navarino Island, southern Chile. The Wilcoxon non-parametric post hoc comparison adjusted with Holm correction is presented in the Appendix A.

	Total Treatments	H	*p*
CT_max_	20	114.09	<0.00001
CT_min_	17	56.272	0.0043
Thermal Breadth	5	92.736	<0.00001

**Table 4 insects-11-00102-t004:** Behavioral responses to water temperature increase (point of thermal reactivity, PTR, and critical thermal endpoint, CTE) of sub-Antarctic macroinvertebrate taxa. Suitability ranks SR: 1 = very suitable, 2 = suitable with limitations, and 3 = not suitable, and thermal sensitivity rank TSR: 1 = very sensitive (≤25 °C), 2 = moderately sensitive (≤30 °C), and 3 = less sensitive (≥30 °C) (adapted from Dallas and Rivers-Moore 2012).

Macroinvertebrate Taxa	Behavioral Response	SR	TSR
Amphipoda: Hyalellidae *Hyalella* sp.	Before temperature increase, individuals swam intermittently in the experimental chambers, moving from bottom to top, and vice-versa. The PTR was apparent as a substantial decrease in swimming speed. The CTE was identified when individuals remained at the bottom of the chamber and stopped swimming. The only detectable movement was then from the antennae and hind legs.	2	3
Coleptera: Dytiscidae *Lancetes angusticollis*	Before temperature increase, individuals swam intermittently in the experimental chambers, moving from bottom to top, and vice-versa. The PTR was apparent as a substantial increase in movement, constantly grabbing air bubbles with difficulty in maintaining them. The CTE was identified as inactivity at the bottom of the chamber, or floating near the surface.	3	3
Hemiptera: Corixidae *Sigara* sp.	Before temperature increase, individuals swam intermittently in the experimental chambers, moving from bottom to top, and vice-versa. The PTR was apparent when individuals started to accelerate their movement. The CTE was reached when individuals sank to the bottom of the chamber, or floated at the top with no detectable movement other than the palas (anterior legs).	3	3
Rhabditophora: Tricladida Dugesiidae	Before temperature increase, individuals crawled on the bottom of the chamber. The PTR was apparent when individuals start to agitate their bodies sideways. When the CTE was reached, individuals started to expose their digestive structures outside their bodies and remained almost immobile at the bottom of the chamber.	2	1
Plecoptera: Gripopterygidae *Aubertoperla kuscheli*	Before temperature increase, individuals crawled on the bottom of the chamber, started moving their legs and cerci. As the PTR was reached individuals lost their grip and start swimming slowly. The CTE was identified when individuals lost grip and swimming capacity, floating at the top of the chamber and remaining immobile with the exception of antennal movement.	2	2
Diptera: Chironomidae Tanypodinae	Before temperature increase, individuals crawled on the bottom of the experimental chamber. The PTR was apparent when individuals started to move and shake at the bottom of the chamber. When they reached the CTE, individuals lost the ability to remain attached to the bottom and floated virtually motionless.	1	1

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
