# Peer review of "Sub-Antarctic Freshwater Invertebrate Thermal Tolerances: An Assessment of Critical Thermal Limits and Behavioral Responses"

_insects, 2020, doi:10.3390/insects11020102_

Round 1
Reviewer 1 Report
The present study may be an interesting contribution to the knowledge on thermal tolerances of aquatic macroinvertebrates by providing data for six South American invertebrate taxa. With this study, the authors also intend to test the climate variability hypothesis CVH (as suggested by the conclusions). However, I have some concerns about whether the conclusions drawn from the study results are justified or maybe a bit overambitious. I am not able to evaluate this fully because some important information about experimental conditions is missing, i.e. oxygen supply, size of the test chambers, size and natural altitude range of the test organisms. Apart from this, I think that the statistical analysis has been performed not quite correctly. My main concern, however, is that only six very different taxa were tested, with no relation to the altitude gradient, i.e. no taxa pairs from at least the same family or order were compared. In my opinion, this limits the validity of the conclusions regarding support of the CVH.
The primary goals of the study are clearly stated but the relation to the CVH is not explicitly mentioned at the end of the introduction. As far as I can judge by the given information, the experimental design is not quite perfect due to pseudoreplicates and different numbers of test animals in the chambers (see specific comments). The authors used a traditional and straightforward statistical method that is generally appropriate for this experimental design but they should revise their use of replicates.
The ms is mostly well written and, with some smaller exceptions, easy to read and the figures are meaningful (but their readability could be improved with regard to b/w printing). In cited references, a few important papers should be included. Please see specific comments for details.
Specific comments:
Abstract
L 21 f “climate variability hypothesis” is presented as a fact although its generality is questioned as the authors state in the introduction; it should be rather presented as an assumption. But see my comment on L120ff and L300ff.
Introduction
L35f Sentence is not quit e clear, why are environmental gradients considered “ecological and evolutionary equivalents”?
L48f rather “force” than “allow”?
L51 as temperature was mentioned in the paragraph above already, it does not need to be listed here again (at least not under “also”)
L65f Regardless of whether “temperate and tropical ectotherms” analysed together in the same study or in separate studies is meant in first part of the sentence, I must disagree. There is some literature comparing temperate and tropical organisms directly, although not always exactly with the method used here. See the meta-analysis of Bates et al. 2013 and the book chapter of Saenger & Holmes, 1991(!). Literature is even more abundant regarding data on temperate OR tropical aquatic organisms, which can be shown by a simple Google search with the respective keywords. As the present ms provides data on organisms from “only” one climate zone, giving credit to some of this literature would be appropriate.
Bates AE, McKelvie CM, Sorte CJ, Morley SA, Jones NA, Mondon JA, Bird TJ and Quinn G (2013) Geographical range, heat tolerance and invasion success in aquatic species. Proceedings of the Royal Society B: Biological Sciences 280(1772):20131958.
Saenger P and Holmes N (1991) Physiological, temperature tolerance, and behavioral differences between tropical and temperate organisms. Pollution in Tropical Aquatic Systems.
Rezende EL, Castañeda LE and Santos M (2014) Tolerance landscapes in thermal ecology. Funct Ecol 28(4):799-809.
Methods
L120ff Here, I see a critical point. Why were only six taxa from different orders or even classes or phyla chosen? For low altitude, an amphipod, a beetle and a water bug, and for high altitude, a flatworm, a stonefly larva and a midge larva? The marked phylogenetic, morphological, and physiological differences (example keyword: respiration) between these taxa limit the validity of the conclusions of the study with respect to the “climate variability hypothesis”. Of course, it is OK to include different taxa in one study. However, if the effects of different “treatments” or an environmental gradient are to be compared, more taxa should be analyzed – preferably taxa from families or orders that are common to all treatments should be chosen. Perhaps I am taking a rather strict view but adaptations to environmental factors are related to phylogeny and I think this should be taken into account.
Also, in this context it would be interesting to know the natural altitude range of the tested organisms, i.e. is it broad or narrow, and were the animals collected from sites in the mid or near the borders of their occurrence range? The latter question is not answered alone by high abundances at the sampling site.
L133 “four to six individuals” - in the supplementary data, I see cases with only two individuals per trial.
L134 This makes those 2-6 individuals pseudoreplicates. The individuals from one chamber must not be analyzed separately in the statistical test, as indicated by the sample size n in Table 3, but as a “bulk” (for instance, arithmetic mean). I assume the Kruskal-Wallis test will probably still be significant, as it is very robust, but it should be applied correctly. The same is valid for the regression analysis. The true n is 3 or 4 (number of trials), not 16-24. Using the Kruskal-Wallis rank test with unbalanced designs is partly critically discussed but it should be OK here. A more serious problem could be the possible differences between trials with 2, 4 and 6 individuals in a chamber which might bias the results. How big were the chambers? Were they equipped with refugia to minimize encounters? Can stress due to mutual interference (“crowding”) be ruled out, especially in the highly mobile predatory taxa?
L135 Were oxygen concentrations or at least saturations measured during the experiment? Oxygen being one (or even the most) crucial factor in thermal tolerance, the information that aquaria were aerated is not quite enough, some concrete data should be provided.
Hoefnagel KN and Verberk W (2015) Is the temperature-size rule mediated by oxygen in aquatic ectotherms? Journal of Thermal Biology 54:56-65.
Verberk W, Bilton DT, Calosi P and Spicer JI (2011) Oxygen supply in aquatic ectotherms: Partial pressure and solubility together explain biodiversity and size patterns. Ecology 92(8):1565-1572.
Sprague J (1963) Resistance of four freshwater crustaceans to lethal high temperature and low oxygen. Journal of the Fisheries Board of Canada 20(2):387-415.
L154 “reached”
L170 How was the “ease” quantified or ranked? Did body size play any role in the suitability ranking?
Results
L193 Fig. 2 and L209 Fig 3: to increase readability also in b/w prints, different types of points could be used
L196-207 redundant with Table 2; “angusticollis”; “dugesiid”
L228 Fig 5 to increase readability also in b/w prints, different shadings for the bars could be used
L247 Table 4 Aubertoperla: “started”
Discussion
L249f In my opinion, this was actually not “proven” by this study. But it could be part of the introduction together with the following sentence.
L254 “decrease” or “are lower”.
L296f Size (and age) are briefly discussed here but no concrete size data are given in the methods or results.
L300-306 identical with L309-315. Unfortunately, I cannot see that the present study supports the CVH even with the difference between low and high altitudes may be still significant after correcting the statistical testing method. Please see my comment on L120ff.
Author Response
Cover letter and answers to Reviewer 1 comments and suggestions.
First, we thank reviewer 1 comments and suggestions, as we think they were necessary feedback for our manuscript. We replied to all comments (as follows in point-to-point detail), addressing most of them and discussed the ones we think deserved it. Editors and reviewers will find following this paragraph the detail of each addressed comment, with our reply after “R:”.
Specific comments
L 21 f “climate variability hypothesis” is presented as a fact although its generality is questioned as the authors state in the introduction; it should be rather presented as an assumption. But see my comment on L120ff and L300ff.
R: Commentary addressed.
L35f Sentence is not quit e clear, why are environmental gradients considered “ecological and evolutionary equivalents”?
R: In terms of species adaptations to environmental clines, changes in the text were made.
L48f rather “force” than “allow”?
R: We preferred "allow" because of species with plasticity on thermal limits can fare in new climates (i.e. introduced species; Bates et al. 2013).
L51 as temperature was mentioned in the paragraph above already, it does not need to be listed here again (at least not under “also”).
R: Commentary addressed.
L65f Regardless of whether “temperate and tropical ectotherms” analysed together in the same study or in separate studies is meant in first part of the sentence, I must disagree. There is some literature comparing temperate and tropical organisms directly, although not always exactly with the method used here. See the meta-analysis of Bates et al. 2013 and the book chapter of Saenger & Holmes, 1991(!). Literature is even more abundant regarding data on temperate OR tropical aquatic organisms, which can be shown by a simple Google search with the respective keywords. As the present ms provides data on organisms from “only” one climate zone, giving credit to some of this literature would be appropriate.
Bates AE, McKelvie CM, Sorte CJ, Morley SA, Jones NA, Mondon JA, Bird TJ and Quinn G (2013) Geographical range, heat tolerance and invasion success in aquatic species. Proceedings of the Royal Society B: Biological Sciences 280(1772):20131958.
Saenger P and Holmes N (1991) Physiological, temperature tolerance, and behavioral differences between tropical and temperate organisms. Pollution in Tropical Aquatic Systems.
Rezende EL, Castañeda LE and Santos M (2014) Tolerance landscapes in thermal ecology. Funct Ecol 28(4):799-809.
R: We have changed the emphasis in this sentence and added additional references.
Methods
L120ff Here, I see a critical point. Why were only six taxa from different orders or even classes or phyla chosen? For low altitude, an amphipod, a beetle and a water bug, and for high altitude, a flatworm, a stonefly larva and a midge larva? The marked phylogenetic, morphological, and physiological differences (example keyword: respiration) between these taxa limit the validity of the conclusions of the study with respect to the “climate variability hypothesis”. Of course, it is OK to include different taxa in one study. However, if the effects of different “treatments” or an environmental gradient are to be compared, more taxa should be analyzed – preferably taxa from families or orders that are common to all treatments should be chosen. Perhaps I am taking a rather strict view but adaptations to environmental factors are related to phylogeny and I think this should be taken into account.
R: We do agree with this comment overall. Ideally more and paired taxa are required to give a borader test of the “climate variability hypothesis”. However, taxonomic, phylogenetic and physiological data relating to freshwater fauna in the Magellanic subantarctic ecoregion is scarce, or non-existent, and studies such as the one described are also critically limited by the availability of suitable taxa within the studied lagoons. Our study is part of recent efforts to understand changes in diversity through the elevation gradient and how this relates to environmental factors.
Also, in this context it would be interesting to know the natural altitude range of the tested organisms, i.e. is it broad or narrow, and were the animals collected from sites in the mid or near the borders of their occurrence range? The latter question is not answered alone by high abundances at the sampling site.
R: Natural altitudinal ranges and elevation occurrences of the studied taxa are now given in Table 1 (Table modified). With the exception of diving beetles and amphipods, organisms were collected at mid of their occurrence ranges within the watershed, however we did not considered this aspect. It is an interesting question for assessing differential tolerance of species with broad ranges, or inhabiting several watersheds.
L133 “four to six individuals” - in the supplementary data, I see cases with only two individuals per trial.
R: Sorry, our error, this has been corrected.
L134 This makes those 2-6 individuals pseudoreplicates. The individuals from one chamber must not be analyzed separately in the statistical test, as indicated by the sample size n in Table 3, but as a “bulk” (for instance, arithmetic mean). I assume the Kruskal-Wallis test will probably still be significant, as it is very robust, but it should be applied correctly. The same is valid for the regression analysis. The true n is 3 or 4 (number of trials), not 16-24. Using the Kruskal-Wallis rank test with unbalanced designs is partly critically discussed but it should be OK here. A more serious problem could be the possible differences between trials with 2, 4 and 6 individuals in a chamber which might bias the results. How big were the chambers? Were they equipped with refugia to minimize encounters? Can stress due to mutual interference (“crowding”) be ruled out, especially in the highly mobile predatory taxa?.
R: A Kruskal-Wallis Rank test for unbalanced design was performed, as this analysis accounts for groups with different sample sizes. Pseudoreplication addressed by analizing trials arithmetic means. We also adjusted the Wilcoxon pairwise test with the Holm correction. Experimental chambers were all the same size, and contained subdivisions to rule out encounters, particularly in the predatory beetles. A new figure is included in the methods illustrating the procedure schematically.
L135 Were oxygen concentrations or at least saturations measured during the experiment? Oxygen being one (or even the most) crucial factor in thermal tolerance, the information that aquaria were aerated is not quite enough, some concrete data should be provided.
R: Minimum oxygen saturation values are now included and referenced.
Hoefnagel KN and Verberk W (2015) Is the temperature-size rule mediated by oxygen in aquatic ectotherms? Journal of Thermal Biology 54:56-65.
Verberk W, Bilton DT, Calosi P and Spicer JI (2011) Oxygen supply in aquatic ectotherms: Partial pressure and solubility together explain biodiversity and size patterns. Ecology 92(8):1565-1572.
Sprague J (1963) Resistance of four freshwater crustaceans to lethal high temperature and low oxygen. Journal of the Fisheries Board of Canada 20(2):387-415.
L154 “reached”.
R: Corrected.
L170 How was the “ease” quantified or ranked? Did body size play any role in the suitability ranking?
R: Based on three features: behavioral responses at PTR and CTE, size/age, and availability in collection sites. We agree that body size is a feature that could alter results, and therefore collection of a wide range of sizes was avoided, and we only tested individuals of similar size/ age or developmental stage.
Results
L193 Fig. 2 and L209 Fig 3: to increase readability also in b/w prints, different types of points could be used.
R: Commentary addressed.
L196-207 redundant with Table 2; “angusticollis”; “dugesiid”.
R: Corrected.
L228 Fig 5 to increase readability also in b/w prints, different shadings for the bars could be used
R: Commentary addressed.
L247 Table 4 Aubertoperla: “started.
R: Corrected.
Discussion
L249f In my opinion, this was actually not “proven” by this study. But it could be part of the introduction together with the following sentence.
L254 “decrease” or “are lower”.
R: Corrected.
L296f Size (and age) are briefly discussed here but no concrete size data are given in the methods or results.
R: Commentary addressed.
L300-306 identical with L309-315. Unfortunately, I cannot see that the present study supports the CVH even with the difference between low and high altitudes may be still significant after correcting the statistical testing method. Please see my comment on L120ff.
R: The results obtained are consistent with some predictions of Janzen´s CVH, more climate variable environments may select for broad thermal tolerant taxa, while less variable environments select for narrower thermal tolerant taxa. We agree that more robust test of this hypothesis is needed, and they require more taxa, comparisons among relative taxa (family or genus level), as well as testing mechanisms of thermal tolerance variation, such as acclimation, intensity and duration of thermal stress.

Reviewer 2 Report
Comments on Carcamo et al.
This paper conducts an experimental study to gain knowledge about the thermal tolerances of freshwater invertebrate species living at different altitudes and thus experiencing different thermal regimes. The authors hypothesized, in line with theory, that species leaving in more variable climates would display larger thermal tolerances than species living in more stable climatic conditions. Their results are in line with this hypothesis. Overall, I found this study interesting and well designed. I of course have some comments.
Major comments:
I’m not familiar with experimental manipulations and I’m therefore also not familiar with the Critical Thermal Method. I assume that this method has already proved useful in other studies but I’m not sure and some references would help in this regard. In particular, I’m wondering whether some papers have compared the results obtained with this method to the results obtained with lethal methods where that critical thermal maximum (or minimum) is determined when individuals are dead. Adding a reference on this would make the use of this method more convincing to me. Currently, it seems that thermal limits are subjectively determined based the observer depending on the perceived behavior of invertebrates. The thing is that I don’t know if the modification in behavior is obvious or not and whether this change would differ depending on the observer.
It would be nice to have a figure summarizing the different steps followed to assess CTmax and CTmin. It took me quite some time and multiple readings to figure out what has been done and even after that, I’m not fully sure that I captured the entire procedure very well.
Control is missing in the experiments. I don’t know if this would affect the results but when applying an experimental, it’s always advised to conduct the same experiment but without the treatment to ensure that the observed response is not just due to random processes (or to the acclimation phase). Would it be possible to add a control?
Minor comments:
The English can be improved here and there. Below are a few examples but others may remain throughout the MS.
Line 32: through utilizing -> along
Line 35: ecological -> ecologically
Line 52: amongst -> among
Line 77: of current study -> of the current study
Line 147: carried -> carried out
Line 254. CTmax lower -> CTmax was lower
Line 119. Can you please briefly explain what the PTR and CTE are (i.e. what do they represent)?
Line 122. Why did you pool together the medium and high altitude sites? Is it because of the low sample size in each? Keeping the medium and high altitude sites separate would have make it possible to potentially highlight a gradient in thermal tolerances, and to provide further support for the hypothesis.
Line 126. “invertebrates were kept at 5 ± 1 °C (approximate water temperature at collection site)”. Does it mean that water temperature wass the same at the three sites at the time of collection? I would have expected that individuals collected at the different sites would be kept at different temperatures matching with the temperature of the sites where they have been collected. Could this acclimation period have an influence on the results?
Lines 139-141. It would be nice to define the behavioral responses. What are the expected responses as the temperature changes? How do you detect these responses? When saying “when an individual exhibited signs of reaching its CTE”; what are these signs? Some elements are provided in Table 4 but it would be helpful to bring those earlier in the MS and to better describe how the behavioral responses were characterized/identified.
Line 145. Is there any reason why 50% individuals are considered to stop the experiment? I’m just wondering since no references are provided.
Line 158-160. “Only individuals that recovered and exhibited normal locomotive functions (antennal, gill, leg, or body movement) were included in the analyses”. Why is this? Is it the normal procedure? Again a reference would help.
Lines 169-173. What do these scales mean? Could you be more specific? For instance I’m not sure to understand what “very adequate” means?
Line 179. Were the Wilcoxon rank tests adjusted for multiple comparisons using e.g. Holm correction?
Line 190. How is it possible that min and max temperature (with more than 15°C difference) be recorded the same month (February 2016). A problem with the data logger?
Line 198. Latin name should be italicized.
Line 200. Close the bracket; remove “respectively”.
Line 214. Linear regressions are not mentioned in the methods.
Line 221. How was the range calculated at the study sites? Is it the difference between the max and min observed values (without averaging). The range should be calculated without averaging.
Table 3. The p-values can be rounded at two decimals.
The conclusion is a large repeat of the last paragraph of the discussion. I suggest to delete this last paragraph and only conserve the conclusion.
Author Response
Cover letter and answers to Reviewer 2 comments and suggestions.
First, we thank reviewer 1 comments and suggestions, as we think they were necessary feedback for our manuscript, particularly in methods and results section. We replied to all comments (as follows in point-to-point detail), addressing most of them and discussed the ones we think deserved it. Editors and reviewers will find following this paragraph the detail of each addressed comment, with our reply after “R:”.
Mayor comments
Commentary 1: I’m not familiar with experimental manipulations and I’m therefore also not familiar with the Critical Thermal Method. I assume that this method has already proved useful in other studies but I’m not sure and some references would help in this regard. In particular, I’m wondering whether some papers have compared the results obtained with this method to the results obtained with lethal methods where that critical thermal maximum (or minimum) is determined when individuals are dead. Adding a reference on this would make the use of this method more convincing to me. Currently, it seems that thermal limits are subjectively determined based the observer depending on the perceived behavior of invertebrates. The thing is that I don’t know if the modification in behavior is obvious or not and whether this change would differ depending on the observer.
R: More references on Critical Thermal Methods use are given, and also a reference of a comparison between lethal and dynamic methods. A reference and statement on how behavior responses were assessed are also given in methods section to address reviewer suggestion. In order to address modification in behavior, we accounted for body or body parts movements, based on taxa not included in this work and literature (see Dallas & Rivers-Moore 2012, now referenced in this section). We observed individuals while in aquaria to assess “pre-treatment” behavior, “pre-treatment in chamber” before trial (detailed in Table 4 Results section), and “treatment” behavior was particularly based on organism general movement (i.e. midges and flatworms) or specific body parts movement (i.e. antennae, legs). By accounting this responses, we avoided misinterpretation of behavior responses.
Commentary 2: It would be nice to have a figure summarizing the different steps followed to assess CTmax and CTmin. It took me quite some time and multiple readings to figure out what has been done and even after that, I’m not fully sure that I captured the entire procedure very well.
R: A new figure that summarize experimental procedure is now included in the MS as Figure 2. Remaining figures are re-numbered accordingly.
Commentary 3: Control is missing in the experiments. I don’t know if this would affect the results but when applying an experimental, it’s always advised to conduct the same experiment but without the treatment to ensure that the observed response is not just due to random processes (or to the acclimation phase). Would it be possible to add a control?
Response 3: We couldn´t address this commentary.
Minor comments
Line 32: through utilizing -> along
R: Commentary addressed.
Line 35: ecological -> ecologically
R: Commentary addressed.
Line 52: amongst -> among
R: Commentary addressed.
Line 77: of current study -> of the current study
R: Commentary addressed.
Line 147: carried -> carried out
R: Commentary addressed.
Line 254. CTmax lower -> CTmax was lower
R: Commentary addressed.
Line 119. Can you please briefly explain what the PTR and CTE are (i.e. what do they represent)?
R: A brief description of PTR and CTE are now included in the introduction.
Line 122. Why did you pool together the medium and high altitude sites? Is it because of the low sample size in each? Keeping the medium and high altitude sites separate would have make it possible to potentially highlight a gradient in thermal tolerances, and to provide further support for the hypothesis.
R: At first we opted to pool medium and high altitude because of low sample size, now we opted to separate them, thus highlight our results and hypothesis support.
Line 126. “invertebrates were kept at 5 ± 1 °C (approximate water temperature at collection site)”. Does it mean that water temperature wass the same at the three sites at the time of collection? I would have expected that individuals collected at the different sites would be kept at different temperatures matching with the temperature of the sites where they have been collected. Could this acclimation period have an influence on the results?
R: We collected organisms belonging to different elevation in different dates. Starting from high elevation downwards. This procedure is based on the marked temperature gradient associated to Magellanic subantarctic freshwater systems (Contador et al. 2015). Also, early to mid autumm sampling was performed before snow season (Late May/ early June) and sampling sites become inaccessible. Temperature was recorded at each sampling event with a multimeter sensor (now included in methods), and then compared to logger data to have a consistent temperature record.
Lines 139-141. It would be nice to define the behavioral responses. What are the expected responses as the temperature changes? How do you detect these responses? When saying “when an individual exhibited signs of reaching its CTE”; what are these signs? Some elements are provided in Table 4 but it would be helpful to bring those earlier in the MS and to better describe how the behavioral responses were characterized/identified.
R: Definition of behavioral responses and expected responses are now included in this section. PTR and CTE signs are now included in this section.
Line 145. Is there any reason why 50% individuals are considered to stop the experiment? I’m just wondering since no references are provided.
R: Reference provided.
Line 158-160. “Only individuals that recovered and exhibited normal locomotive functions (antennal, gill, leg, or body movement) were included in the analyses”. Why is this? Is it the normal procedure? Again a reference would help.
R: Reference provided.
Lines 169-173. What do these scales mean? Could you be more specific? For instance I’m not sure to understand what “very adequate” means?
R: Brief descriptions on scales were added to the MS to clarify reviewer comment.
Line 179. Were the Wilcoxon rank tests adjusted for multiple comparisons using e.g. Holm correction?
R: Holm correction for Wilcoxon post hoc comparisons added in methods and results.
Line 190. How is it possible that min and max temperature (with more than 15°C difference) be recorded the same month (February 2016). A problem with the data logger?
R: This could be an effect of lagoon exposure rather than logger problems.
Line 198. Latin name should be italicized.
R: Commentary addressed.
Line 200. Close the bracket; remove “respectively”.
R: Commentary addressed.
Line 214. Linear regressions are not mentioned in the methods
R: Commentary addressed.
Line 221. How was the range calculated at the study sites? Is it the difference between the max and min observed values (without averaging). The range should be calculated without averaging. R: Ranges were obtained from min and max observed values.
Table 3. The p-values can be rounded at two decimals.
R: Commentary addressed.
The conclusion is a large repeat of the last paragraph of the discussion. I suggest to delete this last paragraph and only conserve the conclusion.
R: Conclusion was improved with reviewers comments, and last paragraph of discussion was deleted.

Round 2
Reviewer 1 Report
The authors have addressed all of my comments I consider important, and improved the ms substantially. I have still some slight reservations concerning the (new) pooled analysis of all six taxa but the results of the corrected statistical analysis seem to justify the procedure. In combination with the now more careful interpretation of the results and additional improvements (e.g. Holm correction, experimental setup scheme) make the ms more comprehensible and also more reproducible and therefore I think it may be published in the present revised form.
Author Response
The authors have addressed all of my comments I consider important, and improved the ms substantially. I have still some slight reservations concerning the (new) pooled analysis of all six taxa but the results of the corrected statistical analysis seem to justify the procedure. In combination with the now more careful interpretation of the results and additional improvements (e.g. Holm correction, experimental setup scheme) make the ms more comprehensible and also more reproducible and therefore I think it may be published in the present revised form.
We are thankful of Reviewer´s comments and suggestions that improved our manuscript.
Reviewer 2 Report
Comments on revised version of Carcamo et al.
Thanks for addressing the comments I provided. I think these changes greatly improved the manuscript. I still have a few comments though.
Line 27. ease in which -> ease with which
Line 34-36. I’m not sure I understand this sentence and the relation with species adaptation. Could you please clarify?
Line 97. “one of its rivers”. Could you be more specific? What ‘its” refers to?
Line 125. So now high elevation is only the site at 700m elevation right? You should thus remove the 480 site within the brackets.
Line 130. I’m sorry but I’m still concerned by the fact that (all?) “invertebrates were kept at 5 ± 1 °C (approximate water temperature at collection site)”. From figure 2 it is clear that temperature is different at the different sites. Thus, putting all individuals at the same temperature likely implies that individuals from some site may not be in the temperature that corresponds to the one where they have been collected, right? If it’s the case, then there is an additional stress for these organisms isn’t it? I’ve no clue to which extent this is the case or not. I’m just noticing that it is a bit strange to say that the same 5°C temperature corresponds to the temperature at the different collection sites. Maybe there is something I misunderstood here.
Regarding my comments about the absence of control, I do understand that these can’t be included in the present study because it would require to perform the experiment and the sampling all over again. I’m not saying that the results are flawed or anything like that due to the missing control. I would however appreciate a discussion of this issue to stress the limit of the study. Experiments must involve a control to ensure that the observed response is due to the effect of the treatment and not to effect of other uncontrolled factors (e.g. oxygen saturation, amount of light, type of aquaria,...). Since the control is missing here, we can’t be 100% sure that the observed response is due to the temperature change (even though it’s probably the case). It would thus be fair to mention that a control is missing and that future studies should include a control to provide stronger evidence that the effect is due to the treatment. This can be done by adding one or two sentences at the beginning of the conclusion section where you provide avenues for future studies.
Line 139. “per trial”. How many trials have you performed?
Line 147-148. “When an individual exhibited signs of reaching its CTE”. Please give an example “(e.g. )”. You can maybe also refer to Table 4 where the behavioural responses are listed for the different taxa.
Line 188. Lineal -> linear
In Figure 3, it would be nice to add an arrow above each curve to indicate when the sampling has been done.
Figure 4. Why not putting all the taxa on the same figure (i.e. not have two panels but just one). This would make the comparison between taxa easier. To keep the information about low, mid and high elevation, you can use three different colors with different shades for the taxa belonging to the same elevation site. You would thus have e.g. light green, normal green and dark green for low elevation taxa, blue for the mid elevation taxa and light red and normal red for the high elevation taxa. What do the vertical bars (SD) represent? Is it the uncertainty in the CTmax assessment (the errors reported in Table 2)? In that case, they should be horizontal.
Line 230. “taxa” should be outside of the brackets.
Author Response
Cover letter and answers to Reviewer 1 comments and suggestions.
First, we thank again reviewer 1 for pointing out some relevant details, comments and suggestions. We replied to all comments as follows in point-to-point detail. Editors and reviewers will find following this paragraph the detail of each addressed comment, with our reply after “R:”.
Line 27. ease in which -> ease with which
R: Corrected.
Line 34-36. I’m not sure I understand this sentence and the relation with species adaptation. Could you please clarify?
R: Latitudinal and altitudinal gradients generally influence environmental features in analogous ways, higher latitudes will be colder just like high elevations. These spatial and environmental gradients will select for physiological traits that can be considered the driving force of species ranges and boundaries, or a by-product of local adaptation (Bozinovic et al. 2011).
Line 97. “one of its rivers”. Could you be more specific? What ‘its” refers to?
R: Sentence reworked. “its” refers to one of Navarino Island´s rivers.
Line 125. So now high elevation is only the site at 700m elevation right? You should thus remove the 480 site within the brackets.
R: Corrected.
Line 130. I’m sorry but I’m still concerned by the fact that (all?) “invertebrates were kept at 5 ± 1 °C (approximate water temperature at collection site)”. From figure 2 it is clear that temperature is different at the different sites. Thus, putting all individuals at the same temperature likely implies that individuals from some site may not be in the temperature that corresponds to the one where they have been collected, right? If it’s the case, then there is an additional stress for these organisms isn’t it? I’ve no clue to which extent this is the case or not. I’m just noticing that it is a bit strange to say that the same 5°C temperature corresponds to the temperature at the different collection sites. Maybe there is something I misunderstood here.
R: We added a sentence in order to clarify this issue: “Organisms inhabiting medium and high elevation were collected in March because of accessibility to sampling sites, while lower elevation site was surveyed in April. The main reason of this sampling lapse is the accessibility to sites and prevalent harsh weather conditions”. “High” elevations of this ecoregion (+400 m a.s.l.) are hard to sample from from April to October, accessibility is limited by sleet and strong winds, and by late April snow starts to accumulate at these sites.
Regarding my comments about the absence of control, I do understand that these can’t be included in the present study because it would require to perform the experiment and the sampling all over again. I’m not saying that the results are flawed or anything like that due to the missing control. I would however appreciate a discussion of this issue to stress the limit of the study. Experiments must involve a control to ensure that the observed response is due to the effect of the treatment and not to effect of other uncontrolled factors (e.g. oxygen saturation, amount of light, type of aquaria,...). Since the control is missing here, we can’t be 100% sure that the observed response is due to the temperature change (even though it’s probably the case). It would thus be fair to mention that a control is missing and that future studies should include a control to provide stronger evidence that the effect is due to the treatment. This can be done by adding one or two sentences at the beginning of the conclusion section where you provide avenues for future studies.
R: In addition to Editor´s comments on this issue, we added a sentence to clarify the “absence” of control. The 24 h at constant temperature (5 °C) prior to experimental procedures could be considered a surrogate of control, at least in some respects. We added detailed description of this in methods section, and following your suggestion we added a sentence in conclusion section.
Line 139. “per trial”. How many trials have you performed?
R: Corrected.
Line 147-148. “When an individual exhibited signs of reaching its CTE”. Please give an example “(e.g. )”. You can maybe also refer to Table 4 where the behavioural responses are listed for the different taxa.
R: Example given, and Table 4 referenced.
Line 188. Lineal -> linear
R: Corrected
In Figure 3, it would be nice to add an arrow above each curve to indicate when the sampling has been done.
R: Arrows added to Figure 3 and explanation in figure text.
Figure 4. Why not putting all the taxa on the same figure (i.e. not have two panels but just one). This would make the comparison between taxa easier. To keep the information about low, mid and high elevation, you can use three different colors with different shades for the taxa belonging to the same elevation site. You would thus have e.g. light green, normal green and dark green for low elevation taxa, blue for the mid elevation taxa and light red and normal red for the high elevation taxa. What do the vertical bars (SD) represent? Is it the uncertainty in the CTmax assessment (the errors reported in Table 2)? In that case, they should be horizontal.
R: Figure arranged as suggested.
Line 230. “taxa” should be outside of the brackets.
R: Corrected.
